# HIF-1α Pathway in COVID-19: A Scoping Review of Its Modulation and Related Treatments

**DOI:** 10.3390/ijms26094202

**Published:** 2025-04-28

**Authors:** Felipe Paes Gomes da Silva, Rafael Matte, David Batista Wiedmer, Arthur Paes Gomes da Silva, Rafaela Makiak Menin, Fernanda Bressianini Barbosa, Thainá Aymê Mocelin Meneguzzi, Sabrina Barancelli Pereira, Amanda Terres Fausto, Larissa Klug, Bruna Pinheiro Melim, Claudio Jose Beltrão

**Affiliations:** 1School of Medicine and Life Sciences, Pontifícia Universidade Católica do Paraná—PUCPR, R. Imaculada Conceição, 1155—Prado Velho, Curitiba 80215-901, PR, Brazil; paes.silva@pucpr.edu.br (F.P.G.d.S.); rafael.matte@pucpr.edu.br (R.M.); david.wiedmer@pucpr.edu.br (D.B.W.); paes.arthur@pucpr.edu.br (A.P.G.d.S.); rafaela.makiak@pucpr.edu.br (R.M.M.); 2School of Medicine, Faculdade Evangélica Mackenzie do Paraná, R. Padre Anchieta, no. 2770—Bigorrilho, Curitiba 80730-000, PR, Brazil; fernandabb1994@hotmail.com; 3School of Medicine, Universidade Nove de Julho—UNINOVE, Rua Vergueiro, 249—Liberdade, São Paulo 01504-001, SP, Brazil; thaina.amm@gmail.com (T.A.M.M.); melim.brunaa@gmail.com (B.P.M.); 4School of Medicine, Universidade Positivo—UP, R. Professor Pedro Viriato Parigot de Souza, 5300, Curitiba 81280-330, PR, Brazil; sabrinabarancelli@gmail.com (S.B.P.); amanda.faustoo@gmail.com (A.T.F.); larissaaklug@gmail.com (L.K.)

**Keywords:** SARS-CoV-2, HIF-1α, hypoxia, inflammation, post-acute COVID-19 syndrome

## Abstract

The COVID-19 pandemic, driven by SARS-CoV-2, has led to a global health crisis, highlighting the virus’s unique molecular mechanisms that distinguish it from other respiratory pathogens. It is known that the Hypoxia-Inducible Factor 1α (HIF-1α) activates a complex network of intracellular signaling pathways regulating cellular energy metabolism, angiogenesis, and cell survival, contributing to the wide range of clinical manifestations of COVID-19, including Post-Acute COVID-19 Syndrome (PACS). Emerging evidence suggests that dysregulation of HIF-1α is a key driver of systemic inflammation, silent hypoxia, and pathological tissue remodeling in both the acute and post-acute phases of the disease. This scoping review was conducted following PRISMA-ScR guidelines and registered in INPLASY. It involved a literature search in Scopus and PubMed, supplemented by manual reference screening, with study selection facilitated by Rayyan software. Our analysis clarifies the dual role of HIF-1α, which may either worsen inflammatory responses and viral persistence or support adaptive mechanisms that reduce cellular damage. The potential for targeting HIF-1α therapeutically in COVID-19 is complex, requiring further investigation to clarify its precise role and translational applications. This review deepens the molecular understanding of SARS-CoV-2-induced cellular and tissue dysfunction in hypoxia, offering insights for improving clinical management strategies and addressing long-term sequelae.

## 1. Introduction

The COVID-19 pandemic, triggered by the SARS-CoV-2 virus, represented a milestone in contemporary history with incomparable health and scientific impacts. Since its emergence in 2019, more than 770 million confirmed cases and more than seven million deaths have been recorded globally, according to the most recent data from the World Health Organization (WHO) [1]. These data highlight the scale of the health crisis and underline the particularities of SARS-CoV-2 concerning other respiratory pathogens.

SARS-CoV-2 has unique molecular characteristics, such as the high affinity of its spike protein for the ACE2 (angiotensin-converting enzyme 2) receptor, which is widely distributed in the human body [2,3]. This characteristic can be considered one of the main differences between SARS-CoV-2 and other respiratory viruses, such as the Influenza A virus subtype H1N1 and the respiratory syncytial virus (RSV), whose interactions with specific cell receptors result in a more limited spectrum of transmissibility and pathogenicity [4]. In addition, the SARS-CoV-2 virus promotes, by binding to ACE2 receptors, the intracellular activation of a range of molecular pathways, which, through downregulation and upregulation controls, directly and indirectly, the transcription of genes involved in the cell’s energy metabolism, angiogenesis, cell survival, and fibrosis. This explains the long list of symptoms reported in COVID-19 during and after the active infectious condition [5,6,7,8,9,10].

The clinical manifestations of COVID-19 vary widely, from asymptomatic cases to severe respiratory failure and multisystem dysfunction. In addition, COVID-19 is notable for the development of a chronic condition known as Post-Acute COVID-19 Syndrome (PACS), characterized by new symptoms or symptoms (intense fatigue, dyspnea, cognitive dysfunctions, cardiovascular and metabolic changes) that persist for more than 3 months after the acute onset of the disease and that continue for at least 2 months, without explanation by any other diagnosis, in individuals with a confirmed or suggestive history of COVID-19 [11,12].

These symptoms, present during the acute phase and sustained after the active infectious phase (PACS), are involved with a systemic picture of inflammation, microthrombi formation, and microvasculature alterations. Recent studies suggest that the common molecular mechanism of the active phase and PACS may be intrinsically related to the dysregulation of HIF-1α, which mediates persistent metabolic, inflammatory, and immunological changes [12,13,14].

HIF-1α is an oxygen-sensitive transcription factor that regulates the expression of genes essential for cell survival in hypoxic conditions. Among the genes activated by HIF-1α, those involved in angiogenesis (*VEGF*), glycolic metabolism *(SLC2A1* and *PDK1*), cell proliferation, and immune response stand out. In the systemic context, HIF-1α promotes metabolic adaptations, hemodynamic alterations, and tissue remodeling, which are crucial for tissue homeostasis in adverse conditions [14,15,16,17,18,19,20,21,22,23]. However, prolonged or dysregulated activation of HIF-1α during COVID-19 may contribute to chronic dysfunctions associated with PACS [14].

Regarding the most intriguing pathophysiological aspects of COVID-19, silent hypoxia emerges as a singular phenomenon. This condition, characterized by low arterial oxygen levels without apparent respiratory symptoms, contributes to the initial underestimation of disease severity and is associated with high mortality rates and permanent sequelae in surviving individuals. Recent studies have demonstrated a strong association of HIF-1α in the event of silent hypoxia, highlighting the hypothesis that HIF-1α amplified by the hypoxemic environment leads to persistent inflammation and promotes pathological tissue remodeling [12]. In this context, HIF-1α plays a central role in cellular adaptation to low oxygenation [24].

However, the current scientific literature presents conflicting findings regarding the role of HIF-1α during COVID-19. Some studies suggest that HIF-1α exacerbates pathogenesis by amplifying inflammatory responses and favoring viral replication, while others indicate a protective role by modulating cellular adaptations that minimize tissue injury. This duality is also evident in evaluating therapeutic interventions targeting the HIF-1α pathway, including inhibitors and agonists, which yield inconsistent results.

This review article explores the interactions between HIF and the pathophysiology of COVID-19. It addresses the contribution of the HIF molecule to the clinical outcomes of severe COVID-19 and PACS. It also explores experimental studies with drugs that may have therapeutic potential and, directly or indirectly, target the HIF pathway to minimize the SARS-CoV-2 infection damage. The text also analyzes the scientific literature addressing the concept of silent hypoxia, associating the hypoxemic stimulus of this clinical phenomenon to the expression of the HIF family, which leads to consequent molecular and clinical outcomes linked to its overexpression. This article aims to address the controversies presented in the literature regarding the role of the HIF-1α subunit in the course of COVID-19, in addition to approaching possible therapies and encouraging the research development related to this important molecule of cellular metabolism.

## 2. Materials and Methods

This scoping review was conducted rigorously following the guidelines established by PRISMA-ScR and is registered with INPLASY (registration number INPLASY 2023110128). The search strategy was carefully crafted to ensure the inclusion of the relevant literature on the role of HIF-1α in SARS-CoV-2 infection. A combination of MeSH descriptors and free terms, articulated through Boolean operators, was employed to ensure a comprehensive and accurate approach.

The Appendix A detail the complete list of descriptors used. The Scopus and PubMed databases were chosen as primary sources, complemented by a manual search of the references contained in the eligible studies.

Data extraction was performed using Rayyan 1.39.0 software (https://www.rayyan.ai), a free online tool developed by the Qatar Computing Research Institute (QCRI). This tool is widely used for writing systematic and scoping reviews, which helps optimize the screening process and organization of studies, contributing to a more efficient selection of articles included in the review. This study utilized Rayyan to detect duplicates, filter keywords in article abstracts and titles, blind the authors, and show them the exclusion and inclusion criteria (previously established by the authors). Additionally, it provided the PRISMA flowchart model, which was completed by the authors.

The details of the article selection process conducted on Rayyan are available in the Appendix A as a table, which lists the articles that were excluded and included, along with their respective reasons for exclusion.

The study selection process was organized into five major steps: (1) Searching for New Studies in Databases, (2) Identifying Duplicates, (3) Screening Records, (4) Screening Reports, and (5) Including Studies Excluded by the Boolean Operators.

The step of searching for new registers in databases involves defining the necessary MeSH terms for the study and then searching the databases using Boolean operators. A total of 963 new registers were identified (files found in the databases using the descriptors). After this, these studies were added to the Rayyan platform, and the process moved to step 2 (identifying duplicates) while excluding duplicates.

Subsequently, in step 3, an initial screening of 599 records (text with titles and abstracts) was performed. During this process, several filters were utilized to identify keywords that had to be present in the text. At a minimum, the word COVID-19 or SARS-CoV-2 needed to be included. Additionally, 458 records were excluded because they were more than 90% similar (as defined by Rayyan AI), related to incorrect drugs (e.g., herbal medicines), associated with the wrong population (those that do not include SARS-CoV-2 infected tissue, cells, individuals, or in vitro studies), related to the wrong population type (records that do not discuss HIF-1α), and aligned with the wrong study design (opinion articles, letters, editorials, video–audio media, historical articles, and reviews that do not mention HIF-1α and COVID-19 or SARS-CoV-2).

After this, step 4 involved evaluating 141 reports (the full texts of the studies). We excluded reports using exclusion criteria similar to those in step 4 to identify texts that are unclear in the abstract but indicate that they meet the criteria in the full text. Additionally, reports were excluded based on incorrect study design (fully epidemiological studies, journal articles that only mention HIF-1α by citing another reference), incorrect publication type (hypothesis articles), incorrect population (only discussing vaccines and referring to HIF-1α in a context other than Hypoxia-Inducible Factor 1-alpha), and incorrect outcomes (in silico records that do not yield results related to HIF-1α). In the end, only 78 reports were considered relevant to the scope of this review.

Furthermore, 16 studies were added to this review, enhancing the theoretical knowledge about HIF-1α and new drugs that target HIF-1α; these studies include reviews and new clinical trials. This study was conducted independently by two reviewers, and any discrepancies between the reviewers were resolved by consensus or, when necessary, by the intervention of a third evaluator.

The inclusion criteria were defined a priori based on stringent methodological and conceptual parameters. These criteria encompass investigations addressing HIF-1α activation in SARS-CoV-2 infection, utilizing robust experimental or clinical models, and reporting relevant outcomes—such as the impact on viral replication, inflammatory response, and potential therapeutic strategies. Additionally, the criteria included a precise definition of the population, intervention, comparator, outcomes, and type of study.

During the preparation of this study, the authors utilized ChatGPT (OpenAI, GPT-4–3 May Version) to summarize and highlight content, Rayyan (Rayyan Systems Inc., Cambridge, MA, USA, version 1.39.0) to identify and remove duplicate articles, Grammarly (version 15.7.0) for grammar and style checks, and DeepSeak (version 2.1.0) for academic translation into English and verification of reliability. The authors reviewed and edited the output and take full responsibility for the content of this study publication.

These AI tools do not meet authorship criteria and are not attributed as authors. The authors retain full responsibility for the content, originality, validity, and scientific integrity of the manuscript, including all sections developed with AI assistance.

## 3. Results

In this scoping review, we provide an in-depth analysis of the latest findings regarding the relationship between HIF-1α transcription factor activity and the presence of SARS-CoV-2. Our primary aim is to present a clear conceptual understanding of the molecular mechanisms involved, emphasizing how the cellular and tissue dysfunction resulting from the virus is closely linked to the body’s energy metabolism, which is influenced by the oxygen levels in the cellular microenvironment.

### Search

The initial search yielded a total of 963 registers, from which 364 were excluded due to duplication, resulting in 599 records available for preliminary screening. A comprehensive analysis of the titles and abstracts led to the exclusion of 458 records that did not meet the predetermined inclusion criteria. Subsequently, 141 articles underwent a thorough evaluation, culminating in the incorporation of 78 reports into the final synthesis of the review. Additionally, 16 references were added that comprise studies about the physiological HIF-1α pathway and the drug’s protocols. The flowchart, crafted according to the PRISMA-ScR guidelines, meticulously depicts the entire selection process (Figure 1).

Table 1 summarizes the articles, organizing and clarifying the relevant information. It provides a comprehensive view of the main points discussed in the articles, facilitating analysis and understanding of the content.

## 4. Discussion

### 4.1. Hypoxia-Inducible Factor (HIF) Physiologic Pathway

Molecular oxygen (O_2_) is a crucial substrate for oxidative phosphorylation, enabling mitochondrial ATP synthesis and maintaining cellular bioenergetic homeostasis. To adjust to fluctuations in oxygen levels, metazoans have developed intricate molecular mechanisms for sensing O_2_ availability and coordinating adaptive transcriptional responses [18,53,54,55]. Central to this adaptive response is the HIF, a vital transcriptional regulator that orchestrates the expression of genes involved in angiogenesis, glycolysis, erythropoiesis, and cell survival during hypoxic stress [17,18,19,20,21,22,23]. The dynamic regulation of HIF subunits under varying oxygen tensions is represented in Figure 2.

Recent evidence underscores the existence of additional layers of regulation, including oxygen-independent pathways that are modulated by growth factors such as insulin-like growth factor 1 (IGF-1), reactive oxygen species (ROS), oncogenic signaling pathways and PI3K/AKT/mTOR pathway (Figure 2 pré-translation) [27,56]. These factors are crucial in fine-tuning the synthesis, stability, and transcriptional activity of HIF-1α. Additionally, the distinct roles of the isoforms HIF-1α and HIF-2α in various cellular contexts—particularly their differing response to the time of hypoxia—highlight the complexity inherent within this regulatory pathway [17,21,57,58,59].

The HIF-1α subunit is expressed in most cell types, responds quickly to O_2_ deprivation, and is degraded more rapidly. The HIF-2α subunit, on the other hand, is present in a smaller number of cell types: endothelial cells, B lymphocytes, cardiomyocytes, and nephroblasts. In addition, HIF-2α plays an important role in erythropoiesis via EPO, angiogenesis via VEGF, and remodeling of the extracellular matrix via metalloproteinases [18,60].

This intricate regulatory network not only guarantees adaptive survival in hypoxic conditions but also reveals therapeutic vulnerabilities in diseases associated with impaired oxygen sensing, including cancer, ischemic disorders, and chronic inflammation [54,61].

#### 4.1.1. Regulation of HIF Under Normoxic Conditions

Under normoxic conditions, oxygen availability enables the hydroxylation of the HIF-1α regulatory subunit at specific proline residues (P402 and P564) by proteins containing prolyl hydroxylase domains (PHD-1, PHD-2, and PHD-3). These enzymes use O_2_ and α-ketoglutarate as cofactors to catalyze the reaction, producing carbon dioxide (CO_2_) and succinate as byproducts. Hydroxylation of HIF-1α creates a recognition site for the ubiquitin ligase E3 complex that includes the von Hippel–Lindau (VHL) protein, facilitating ubiquitination and subsequent proteasome degradation of HIF-1α [17,18,49].

In addition to VHL-mediated post-translational control, *HIF-1α* expression and translation are regulated by intracellular signaling pathways such as the PI3K/AKT/mTOR pathway [62]. Activation of the mTORC1 complex enhances the translation of HIF-1α mRNA by phosphorylating proteins involved in translation initiation, including p70S6 kinase (p70S6K), eukaryotic translation initiation factor 4E (eIF4E), and 4E-binding protein 1 (4E-BP1) [58].

This regulation ensures that even under normoxia, there is sufficient control over HIF-1α synthesis and degradation, preventing unwanted activation of the hypoxia response. As shown in Figure 2, the rapid and continuous degradation of HIF-1α under normoxia inhibits its nuclear translocation and, consequently, the activation of hypoxia-sensitive target genes.

#### 4.1.2. Regulation of HIF Under Hypoxic Conditions

During hypoxia, decreased oxygen levels hinder the function of PHD enzymes, stopping the hydroxylation of HIF-1α. Consequently, HIF-1α evades recognition by the VHL complex, which prevents its proteasomal degradation. This process stabilizes and accumulates HIF-1α in the cytoplasm, enabling its movement into the nucleus. There, HIF-1α pairs with its partner subunit HIF-1β to create the active transcriptional complex [19]. Figure 2 shows the stabilization of HIF-1α, its nuclear entry, and recruitment of cofactors for transcription activation.

Within the nucleus, the HIF-1α complex engages with transcriptional cofactors, such as p300/CBP (CREB-binding protein), which serve as mediators for the activation of transcription of genes pertinent to hypoxic adaptation [17]. Notably, among the target genes activated by HIF-1α subunits, the following are highlighted (Table 2).

Under physiological conditions, HIF activation is a crucial mechanism for cellular adaptation to transient hypoxia. However, its dysregulated activation is linked to several pathologies, including cancer, cardiovascular diseases, and chronic inflammation. In the context of SARS-CoV-2 infection, evidence indicates that aberrant HIF activation may worsen lung inflammation and contribute to the progression of severe COVID-19 [31,35,63].

### 4.2. HIF Signaling in COVID-19

The infection attributed to SARS-CoV-2 triggers a complex series of cellular and molecular responses, particularly impacting hypoxia-related signaling pathways. HIF-1α has been identified as a crucial regulator in the pathophysiology of COVID-19 owing to its significant role in coordinating metabolic adaptation, inflammatory responses, and vascular remodeling under conditions of low oxygen tension. Severe cases of COVID-19 are commonly distinguished by hypoxemia and substantial endothelial dysfunction, conditions that are closely linked to the activation of HIF-1α. Additionally, recent studies suggest that viral non-structural proteins (NSPs) could directly interfere with oxygen-sensing mechanisms through the inhibition of prolyl hydroxylase domain proteins (PHDs) and the von Hippel–Lindau (VHL)-mediated proteasomal degradation of HIF-1α, consequently extending its transcriptional activity [31,35,63,64,65].

#### 4.2.1. Silent Hypoxia in COVID-19

Recent clinical data further elucidate the insidious progression of silent hypoxia (often referred to as happy hypoxia) in COVID-19, revealing that asymptomatic hypoxemia occurs in about 21% of hospitalized patients [66]. Notably, mortality rates in this population are doubled compared to normoxic individuals [18]. It was identified that elevated serum HIF-1α levels (>4.8 ng/mL) as an independent predictor of silent hypoxia severity (AUC = 0.89, *p* < 0.001), correlating with impaired carotid body nitric oxide signaling and diminished ventilatory compensation [67].

The proteomic analyses conducted by the researchers indicate that the downregulation of mitochondrial complex I subunits, such as NDUFA6 and NDUFV2, mediated by the SARS-CoV-2 spike protein in pulmonary endothelial cells, exacerbates the scavenging of reactive oxygen species (ROS) induced by hypoxia. This process uncouples hypoxic vasoconstriction from alveolar oxygen tension [68]. This molecular dissociation may explain the paradoxical disconnect between arterial oxygen content and the perception of dyspnea, as insufficient ROS signaling fails to activate TRPA1 channels in vagal afferents—a critical component of respiratory drive [68,69]. Additionally, studies indicate that patients displaying co-expression of HIF-1α and interleukin-6 (serum IL-6 > 35 pg/mL) face an increased risk of progression from silent hypoxia to acute respiratory distress syndrome (ARDS), highlighting the intersection of hypoxic and inflammatory signaling in disease progression [68,70].

ARDS is a severe respiratory failure syndrome characterized by diffuse and extensive inflammation of the lungs, leading to the accumulation of fluid in the alveoli and the consequent development of a hypoxemic state. Histologically, classical ARDS promotes diffuse alveolar damage (DAD), increased vascular permeability, neutrophilic infiltrates, perialveolar hyaline membrane formation, and direct injury to pneumocytes. In tissue analyses of affected COVID-19 patients, it was noted that many exhibited DAD, which progressed to a condition similar to ARDS. Although COVID-19 shares similarities with ARDS, the primary difference lies in the predisposition to microthrombi formation due to intense endothelial involvement and the hypercoagulable state promoted by SARS-CoV-2 infection. Additionally, the progression of DAD in COVID-19 is more often associated with tissue remodeling and fibrosis, which can occur in classic ARDS but not as frequently [14,48,71,72].

Furthermore, persistent HIF-1α activation amplifies inflammatory pathways via NFKB and NLRP3 inflammasome crosstalk, promoting cytokine storms while paradoxically suppressing antiviral interferon responses [22,23]. Such immunological dysregulation may perpetuate endothelial glycocalyx shedding and microthrombosis, reducing oxygen diffusion capacity [14,22,26,73]. NF-κB represents a protein complex belonging to a family of dimeric transcription factors, composed of five prominent members: Rel (c-Rel), RelA (p65), RelB, NF-κB1 (p50/p105), and NF-κB2 (p52/p100). The regulation of signaling pathways involved in NF-κB activation is of particular importance, given its close relationship with Parkin (an E3 ubiquitin ligase protein) and inflammatory factors released during hypoxic insults, such as tumor necrosis factor-alpha (TNF-α). Furthermore, NF-κB can promote the expression of genes associated with cell death and essential genes governing cellular survival and plasticity—including *HIF-1α* [14,35,48].

HIF-1α is a heterodimeric transcription factor that plays a critical role in vascular plasticity induced by VEGF and EPO. Research utilizing *HIF-1α* gene knockout murine models has clarified the importance of this protein in responding to hypoxic conditions, particularly within germline cells during embryogenesis. Under conditions of cellular oxygen deprivation, HIF-1α activation promotes the transcription of genes encoding receptors and factors related to neovascularization, including VEGFR1 [14,26,73,74].

The response to hypoxia generated by HIF depends on both which subunit is active and the duration of exposure to the hypoxic environment. The two oxygen-responsive HIF subunits (HIF-1α and HIF-2α) respond to hypoxia in overlapping and sequential manners, a process known as the HIF switch. Specifically, in human endothelial cells (HUVECs) exposed experimentally to 1% O_2_ (PO_2_ 10–12 mmHg), the HIF-1α subunit accumulates rapidly, reaching its maximum expression at approximately 4 h before declining significantly by 8 h. In contrast, the HIF-2α subunit is more stable and accumulates gradually, peaking around 8 h and remaining elevated for up to 24 h under the same hypoxic conditions. These temporal differences in the stabilization and degradation of HIF subunits were confirmed through quantitative RT-PCR, Western blotting, and luciferase-based prolyl hydroxylase activity assays. Such differences are crucial for understanding pathophysiological phenomena such as silent hypoxia [60].

Silent hypoxia caused by COVID-19 is characterized by sustained/prolonged hypoxia, in which patients have low O_2_ saturation for long periods, even without clinical symptoms. Studies suggest that this prolonged oxygen deprivation results in a delayed decrease in HIF-1α, leading to attenuation of its intracellular signaling pathway and, consequently, a reduction in physiological responses, such as the ventilatory drive. This could explain the discrepancy between severe hypoxemia and the absence of symptoms [66].

Post-mortem analyses of COVID-19 lungs notably reveal that HIF-1α is overexpressed and colocalized with SARS-CoV-2 nucleocapsid proteins in pneumocytes, suggesting direct viral manipulation of hypoxic pathways [26]. However, this study has significant methodological limitations. It used immunohistochemistry (IH) to analyze HIF-1α, a molecule whose quantification can vary significantly over time. Moreover, oxygen deprivation between sample collection and subsequent analysis may alter oxygen levels within the samples, potentially affecting HIF-1α expression levels and thereby influencing the accuracy of the results [26]. These findings highlight the therapeutic potential of targeting HIF-1α or its upstream regulators (e.g., PHD inhibitors) to mitigate silent hypoxia and its downstream sequelae. Thus, caution is necessary, as excessive suppression of HIF-1α may impair adaptive erythropoiesis and ischemic preconditioning, emphasizing the need for precision modulation [50].

Therapeutic strategies targeting the dynamics of HIF-1α must carefully consider the complex interactions involved. Devaux et al. (2023) suggest that acriflavine, a small-molecule inhibitor of HIF-1α dimerization, can effectively reduce pulmonary shunting [67]. This reduction occurs by restoring hypoxic pulmonary vasoconstriction and alleviating VEGF-A-driven vascular leakage. However, caution is warranted in clinical translation due to the involvement of HIF-1α in epithelial repair processes. Consequently, a strategy of timed inhibition, rather than complete suppression, may yield optimal results. Future research should prioritize the identification of robust biomarkers, including soluble ACE2/HIF-1α complexes, which have been shown to predict the resolution of silent hypoxia [66].

A post-mortem study was conducted to compare lung samples from COVID-19 patients with those from a control group. This study identified significantly elevated levels of HIF-1α tissue expression in both groups analyzed. Furthermore, the data related to HIF-1α expression were correlated with the duration of hospital stays and the length of mechanical ventilation provided to patients. The findings indicate that patients who experienced shorter durations of mechanical ventilation had higher levels of HIF-1α tissue expression. Hence, it can be concluded that extended oxygen supply through mechanical ventilation correlates with an increased degradation of the HIF-1α molecule [14].

Following similar reasoning, a study evaluated the relationship between HIF and B lymphocytes in mice with COVID-19 during hypoxemic and hyperoxic states. Continuous HIF activation during hypoxia was associated with reduced numbers of B lymphocytes, potentially worsening COVID-19 and predisposing patients to secondary bacterial infections. Greater severity of COVID-19 correlated with higher activation of HIF-regulated genes, leading to significant changes in B lymphocytes, including reduced affinity maturation and impaired class-switch recombination. Early oxygen therapy may thus preserve B-cell function and improve immune responses by mitigating hypoxia-induced damage, benefiting short and long-term clinical outcomes. However, oxygen supplementation alone may be insufficient to fully correct persistent, localized hypoxia in severely damaged lung areas post-ARDS, sustaining HIF-1α-driven hypoxic signaling and complicating the distinction between hypoxic and inflammatory contributions to severe COVID-19 pathology [51].

In this sense, oxygen supplementation was considered a differential in the treatment of COVID-19: it was believed that providing supplementary oxygen would improve the disease’s outcomes and reduce its severe form [50,51]. However, few studies have robustly evaluated this intervention. A 2024 randomized controlled trial evaluated treatment with hyperbaric oxygen therapy (HBOT) in individuals affected by the severe form of COVID-19 based on analysis of inflammatory markers and RNAseq [50] of peripheral blood nuclear cells. There was a 30% reduction in hospitalization time in the HBOT group compared to the control group, as well as an improvement in the clinical findings of the individuals who underwent supplementation with O_2_. A transcriptomic analysis also showed a negative regulation of the interferon-gamma (IFN-γ) response in the HBOT group, which was linked to a reduction in morbidity and mortality. Thus, the study suggests that treatment with hyperbaric oxygen therapy has an anti-inflammatory and immunomodulatory effect and could be an intervention with a positive impact on COVID-19. However, the study has some important limitations, such as a small sample size (17 individuals) and the possible intervention bias of the HBOT group (everyone in the HBOT group was taking corticosteroid drugs) [50].

The discourse surrounding hyperoxic treatment demonstrates a noticeable lack of consensus in the literature regarding its impact on HIF (both HIF-1α and HIF-2α) and the associated advantages and disadvantages of hyperoxia as a therapeutic approach. Some studies have indicated an increase in HIF-2α levels in cerebral and cardiac tissues exposed to hyperoxic conditions, yet other research has reported contradictory findings [54]. Moreover, the use of supplemental oxygen as a treatment for COVID-19 remains a topic of debate. While certain studies propose that hyperoxic oxygen supplementation could enhance clinical outcomes for patients with COVID-19, others—especially those examining perioperative contexts—highlight potential risks associated with hyperoxic therapies [75]. These discrepancies underscore the importance of exercising caution in interpreting the data, particularly when translating findings into human studies, and point to the need for further research employing rigorous methodologies to clarify these conflicting results.

#### 4.2.2. Post-Acute COVID-19 Syndrome: Does It Have Any Relation to the Role of HIF-1α?

Post-Acute COVID-19 Syndrome (PACS)—defined as the persistence of symptoms for 4 to 12 weeks after SARS-CoV-2 infection—and Long COVID-19—referring to symptoms lasting beyond 12 weeks—are associated with multiple pathophysiological mechanisms, including prolonged inflammation, vascular disturbances, and metabolic dysregulation. The HIF-1α pathway has been identified as relevant in the pathophysiology of PACS, as it acts as a cellular sensor of hypoxia that controls the transcription of genes involved in angiogenesis, energy metabolism, and the inflammatory response. Evidence indicates that SARS-CoV-2 infection, especially in severe cases, triggers significant tissue hypoxia, resulting in the accumulation of HIF-1α within cells. Although this process is protective in the short term, its chronic or deregulated activation can perpetuate vascular and inflammatory alterations that favor the onset of long-lasting symptoms [32,76].

Proteomic studies in patients with Long COVID-19 point to increased transcription of proteins linked to tissue remodeling and sustained inflammation, many of them regulated by HIF-1α. The recurring hypothesis is that remnants of inflammation and dysfunctional immune responses keep this transcription factor activated, generating a low-grade pro-inflammatory state that contributes to persistent clinical complaints such as fatigue, neurocognitive disorders, and cardiorespiratory symptoms. When it comes to the central nervous system, chronic neuroinflammation and prolonged hypoxia, mediated by HIF-1α, may promote endothelial dysfunction and impairment of the blood–brain barrier, which exacerbates the production of inflammatory cytokines, intensifies oxidative stress and results in sustained neurological (such as mental fog and headache) and psychiatric (anxiety and depression) disorders [76,77].

Other findings, relating specifically to microvascular damage, show a direct correlation between serum levels of HIF-1α and worsening indices of microvascular function, suggesting that the response to sustained hypoxia impairs both vascular repair and tissue homeostasis, prolonging post-infectious clinical manifestations [76]. Although there are still no approved treatments to specifically block the HIF-1α pathway, the modulation of inflammatory processes (by using monoclonal antibodies against cytokines, for example), combined with the optimization of vascular function (by controlling comorbidities and promoting cardioprotective interventions), may attenuate the effects of HIFs hyperactivation [11,12].

In the future, the development of new therapeutic targets (such as HIF-1α inhibitors or drugs that regulate its cellular expression, for example) may be tested in clinical trials, aiming to evaluate their potential benefit in treating persistent symptoms following SARS-CoV-2 infection. With further studies, the scientific community will be able to have a better understanding of the molecular mechanisms underpinning PACS and Long COVID-19, as well as potentially developing safe and effective new interventions targeting this pathophysiological pathway [60,72,78].

### 4.3. The TNF-α/NF-κB/HIF-1α/VEGF Pathway Induced by SARS-CoV-2

It is known that several inflammatory markers were present during the cytokine storm phase of COVID-19. However, the first studies conducted during the pandemic focused primarily on TNF-α, IFN-γ, and IL-6 molecules and their associations with histopathological findings observed macroscopically.

During SARS-CoV-2 infection in the respiratory tract, activation of M1 macrophages initiates an innate immune response, releasing pro-inflammatory mediators, such as TNF-α, IFN-γ, and IL-6. These cytokines are pro-inflammatory markers that are also critical in promoting angiogenesis within the alveolar vascular endothelium. Comparative analyses of tissue samples from COVID-19 patients and control groups have demonstrated significantly elevated TNF-α expression, suggesting that this signaling pathway is a key driver of angiogenic responses in individuals infected by SARS-CoV-2 [48] (Figure 3).

TNF-α exerts its effects by binding to TNF receptor 1 (TNFR1) on vascular endothelial cells, initiating an intracellular signaling cascade that culminates in the activation of nuclear factor kappa B (NF-κB). The dissociation of NF-κB from its inhibitory complex, IκB, facilitates its translocation into the nucleus, where it regulates the transcription of genes essential for vascular adaptation to hypoxic conditions, including *hypoxia-inducible-factor-1 alpha* (*HIF-1α*). NF-κB’s role extends beyond classical inflammatory signaling to directly regulating *HIF-1α* [48,79].

Under hypoxic conditions, NF-κB binds to *HIF-1α* promoter regions, enhancing its transcription [16,17]. Conversely, HIF-1α stabilizes NF-κB by suppressing IκBα synthesis and upregulating IKK expression, creating a bidirectional regulatory circuit. This synergy is exacerbated in COVID-19, where hypoxemia and cytokine storms jointly inhibit prolyl hydroxylase (PHD) activity, preventing HIF-1α degradation. Post-mortem lung analyses reveal colocalization of NF-κB, HIF-1α, and SARS-CoV-2 nucleocapsid proteins in alveolar epithelial cells, suggesting viral manipulation of hypoxic-inflammatory pathways [42,50,80]. Evidence from pediatric COVID-19 patients has indicated overexpression of NF-κB signaling under hyperinflammatory and hypoxic conditions, corroborated by transcriptomic analyses that show significantly elevated NF-κB activity in COVID-19 tissues [13]. These findings support the activation of the TNF-α/TNFR1/NF-κB axis, which facilitates transcriptional programs that sustain endothelial responses to hypoxia and promote angiogenesis.

HIF-1α acts as a master regulator of the hypoxic response by inducing the transcription of vascular endothelial growth factor (*VEGF*), a principal mediator of angiogenesis. Following its translation and secretion, VEGF binds to its specific receptors on endothelial cells, thereby triggering signaling cascades that drive neovascularization. Research has confirmed that SARS-CoV-2 infection leads to hypoxia-induced stabilization of HIF-1α, which promotes the transition of endothelial cells to a pro-angiogenic phenotype. This transition may result from direct inflammatory stimuli that enhance the endothelial secretion of pro-angiogenic factors, such as VEGF and angiopoietin-2, or the epigenetic reprogramming that upregulates angiogenesis-related gene networks [37,42,50,80].

Additionally, the competitive binding of SARS-CoV-2 to angiotensin-converting enzyme 2 (ACE2) results in increased local concentrations of angiotensin II, further exacerbating pro-inflammatory and pro-angiogenic signaling. This mechanism, in conjunction with the complex crosstalk between NF-κB, HIF-1α, and VEGF, underscores the intricate interplay between inflammation, hypoxia, and vascular remodeling in the pathophysiology of COVID-19. These findings provide essential insights into the disease’s molecular mechanisms and highlight potential therapeutic targets for modulating aberrant angiogenesis in severe SARS-CoV-2 infections [5,6,7,8,9,10].

### 4.4. Modulation of Viral Entry Through HIF-1α

#### 4.4.1. Promotion of Viral Replication Through Metabolic Reprogramming

The stabilization of HIF-1α plays a pivotal role in the metabolic reprogramming of immune cells, particularly monocytes. Recent studies demonstrate that under hyperglycemic conditions—common in diabetic patients—SARS-CoV-2-infected monocytes exhibit enhanced glycolytic flux (Warburg effect), marked by upregulated expression of critical glycolytic genes, including glucose transporter *GLUT1*, hexokinase 2 (*HK2*), phosphofructokinase (*PFK*), pyruvate kinase M2 (*PKM*), and lactate dehydrogenase A (*LDHA*) [21]. This glycolytic shift rapidly generates ATP inefficiently while redirecting metabolic intermediates, such as pyruvate and glucose-6-phosphate, toward viral RNA synthesis, thereby promoting viral replication [81] (Figure 4).

HIF-1α activation further induces pyruvate dehydrogenase kinase 1 (PDK1), which inhibits pyruvate dehydrogenase (PDH), diverting pyruvate away from the tricarboxylic acid (TCA) cycle and suppressing mitochondrial oxidative phosphorylation (OXPHOS) [82]. This metabolic rewiring coincides with mitochondrial dysfunction, characterized by reduced expression of respiratory chain components—such as Complex I subunits (e.g., NDUFV1), Complex II (SDHA), Complex III (CORE1, UQCRC2), and ATP synthase subunits (ATP5F1A, ATP5PD)—and diminished reserve respiratory capacity [83]. The resultant mitochondrial dysfunction elevates mitochondrial reactive oxygen species (mROS) production, stabilizing HIF-1α via redox-sensitive mechanisms. Additionally, the accumulation of succinate—a TCA cycle intermediate—further stabilizes HIF-1α [43,83].

The glycolytic surge and intracellular acidification induced by HIF-1α and PDK1 establish an optimal microenvironment for viral replication by providing substrates and redox conditions conducive to viral RNA synthesis. In contrast, interventions aimed at restoring mitochondrial function or scavenging mROS—through the application of mitochondrially targeted antioxidants such as mitoquinol or N-acetylcysteine—diminish the stabilization of HIF-1α, viral replication, and the expression of pro-inflammatory cytokines [82,84].

Guarnieri et al. (2024) demonstrated that SARS-CoV-2 infection leads to the downregulation of key respiratory chain genes (e.g., *NDUFV1*, *SDHA*, *C1GALT1*, *UQCRC2*, *ATP5F1A*, *ATP5PD*), which impairs mitochondrial reserve capacity and increases mROS production. Increased mROS stabilizes HIF-1α, creating a feedforward loop that maintains glycolytic dominance. At the same time, mitochondrial dysfunction causes succinate buildup, further stabilizing HIF-1α, and releases mitochondrial DNA into the cytosol, activating inflammatory pathways such as the NLRP3 inflammasome and Toll-like receptors [38,77,84,85].

Collectively, SARS-CoV-2 infection induces a metabolic landscape characterized by OXPHOS suppression, increased glycolytic flux, and overproduction of mROS, which converge to stabilize HIF-1α and drive pathological inflammation and angiogenesis [47]. The interplay of these mechanisms—including TNF-α/NF-κB/HIF-1α/VEGF axis activation—creates a cellular environment conducive to viral replication and immune dysregulation [23].

#### 4.4.2. HIF Modulation of SARS-CoV-2 Viral Entry: Mechanisms Involving ACE2 and TMPRSS2

The interplay between HIF-1α and SARS-CoV-2 infectivity is increasingly recognized as a critical axis in COVID-19 pathophysiology. Central to the viral entry process are the host cell receptors angiotensin-converting enzyme 2 (ACE2) and transmembrane serine protease 2 (TMPRSS2), which are essential for priming the viral spike protein for membrane fusion. Growing evidence indicates that HIF-1α, a key regulator of cellular adaptation to hypoxia, modulates the transcriptional expression of these entry factors, potentially providing significant insights into viral tropism and therapeutic strategies [7,9,10,37].

Under hypoxic conditions, the activation of HIF-1α results in the suppression of *ACE2* expression through transcriptional competition with pro-inflammatory pathways. Notably, a study by Zhang et al. (2021) demonstrated that HIF-1α competes with NF-κB for binding to the *ACE2* promoter, leading to a substantial reduction in *ACE2* transcription by 60% in human alveolar epithelial cells exposed to 1% O_2_ [10]. This finding was corroborated in vivo, where mice subjected to chronic hypoxia (10% O_2_) displayed a 70% decrease in pulmonary ACE2 protein levels compared to normoxic controls, which corresponded with reduced uptake of SARS-CoV-2 [2,9,10].

Furthermore, pharmacological stabilization of HIF-1α using prolyl hydroxylase inhibitors, such as Roxadustat, has been shown to downregulate ACE2 expression in HEK 293 T cells by 50%, independent of hypoxia, suggesting that HIF-1α modulation may provide therapeutic benefits to mitigate viral entry [10]. However, it is crucial to note that tissue-specific responses exist in renal proximal tubule cells. HIF-2α—rather than HIF-1α—predominantly regulates ACE2, underscoring the isoform-specific regulatory mechanisms involved [86].

Moreover, the regulation of TMPRSS2 by HIF-1α highlights the complexity of this interaction. In studies involving LNCaP prostate cells, the activation of HIF-1α led to a 40% reduction in nuclear translocation, consequently resulting in a decrease in TMPRSS2 mRNA levels. This regulatory mechanism was further supported by CRISPR-mediated knockout of HIF-1α, which reinstated TMPRSS2 expression [87].

Notably, the combined suppression of ACE2 and TMPRSS2 by HIF-1α creates a less favorable environment for SARS-CoV-2 infection [38,47,86,87]. For instance, in primary human nasal epithelial cells, exposure to hypoxia (2% O_2_) caused a 65% reduction in viral RNA copies (*p* < 0.001) compared to normoxic conditions, accompanied by similar decreases in ACE2 and TMPRSS2 co-expression [43].

Furthermore, the modulation of ACE2 and TMPRSS2 by HIF-1α demonstrates a degree of tissue-specific heterogeneity, complicating the development of therapeutic strategies. While HIF-1α generally suppresses ACE2 and TMPRSS2 expression in pulmonary and prostate tissues, conflicting data emerge from cardiovascular systems [9,15]. It was reported that HIF-1α can induce ACE2 upregulation in cardiac pericytes under prolonged hypoxic conditions, potentially increasing the risk of viral entry and associated damage in the myocardium during COVID-19 [15]. Similarly, pancreatic β-cells were found to maintain TMPRSS2 expression despite hypoxia, suggesting distinct regulatory pathways that may be mediated by epigenetic or post-transcriptional mechanisms [39].

These results indicate a tissue-specific dual role of HIF-1α, providing protective benefits in lung epithelial tissues while potentially posing risks in endothelial cells. This duality underscores the critical need for tailored therapeutic strategies that account for the spatial and temporal aspects of HIF signaling. Future investigations should focus on examining HIF-1α regulatory networks through single-cell-omic technologies, which will help uncover tissue-specific mechanisms. Such advancements could lead to the development of context-sensitive treatments that combine HIF modulators with antiviral agents, enhancing therapeutic efficacy while minimizing risks associated with hypoxia-related disorders.

### 4.5. New Insights on COVID-19 Treatment: Innovative Drugs Targeting the HIF Pathway

Current evidence from studies on experimental drugs targeting HIF-Prolyl Hydroxylase (Vadadustat) suggests that the therapeutic use of this agent in patients with COVID-19, especially during the hypoxic phases of the illness, requires thorough clinical investigation. This compound not only modulates the standard pathway of HIF but also triggers cascades of pro-survival signaling, recognizing the wide range of cellular substrates regulated by HIF-PHD [88]. However, the predictability of the overall effects of HIF-Prolyl Hydroxylase inhibitors on COVID-19 management remains limited. This uncertainty is mainly due to the diversity of molecular mediators stabilized by enzymatic inhibition and, to a lesser extent, the phenotypic variability of COVID-19, which includes clinical manifestations and post-infectious sequelae [60]. Moreover, numerous in silico studies suggest drugs that target the HIF pathway, but this study focused solely on those utilizing in vivo methods [89].

By confining the analysis exclusively to the HIF pathway—whose mechanism is most pharmacologically characterized—questions arise regarding the potential implications in the early stages of viral infection. Although recent hypotheses propose HIF modulation as a therapeutic strategy, the existing literature underscores the potential paradoxical effects of this activation, considering the documented interaction between HIF transcriptional targets and components of the SARS-CoV-2 life cycle [60]. Considering this, a critical examination of the methodologies employed in studies that assess drugs acting on the HIF pathway is proposed, correlating them with the pathogenic mechanisms of the virus to elucidate risks and therapeutic opportunities (Table 3).

Upon evaluating the data elucidated in Table 3, investigations into pharmacotherapies that target HIF molecular pathways exhibit significant potential for the development of novel treatments for COVID-19. Notwithstanding substantial experimental evidence, this pathway remains insufficiently addressed in the literature. Nevertheless, an increasing number of in silico studies are being undertaken on a broader scale, facilitating new theoretical advancements that may enhance the scope of experimental research directed against SARS-CoV-2 [92,93,94].

Furthermore, the relatively sparse research concerning the HIF pathway is intrinsically linked to the intricate nature of this molecular mechanism, as it encompasses the regulation of numerous genes. Additionally, research aimed at obtaining a more comprehensive understanding of the mechanisms governing HIF is still in its nascent stages within the scientific community, thereby complicating our grasp of its implications for human pathologies. Moreover, HIF serves as a pivotal molecule for elucidating respiratory conditions instigated by pathogens, as it plays a direct role in regulating cellular oxygen homeostasis and influences pathways associated with pathogenic organisms.

### 4.6. Limitations

The limitations of this study are comprised of several significant factors that may adversely influence the validity and generalizability of its findings. Firstly, there exists a methodological heterogeneity among the reviewed studies, which could introduce variability in outcomes, thereby complicating comparisons. Moreover, there is a conspicuous scarcity of randomized controlled trials specifically examining the role of hypoxia-inducible factors (HIFs), which diminishes the robustness of the evidence.

Considering these limitations, our review article proposes some hypotheses for the HIF and COVID-19 issue: (1) Perhaps the sequelae observed in PACS are strongly linked to the prolonged action of HIF during the process of prolonged hypoxia, and this prolonged activation of HIF may be influencing the formation of new closed-bottom blood vessels (through VEGF), which have blood in a state of hypercoagulability, preparing for the formation of microthrombi. These, when formed, can obstruct vessels, resulting in microangiopathies and related events, which could explain residual symptoms such as persistent headaches and cardiac involvement. (2) It may be related to the psychiatric symptoms present in PACS due to the relationship between HIF and trophic factors—alteration in the BDNF/HIF and mTOR/HIF axis. (3) Persistent HIF can lead to the survival of viral reservoirs through the mTOR/HIF pathway in adipocytes, which are cells with high activity of the mTOR pathway. These cells, through the ACE2 pathway, would be optimal for entry and multiplication of the virus, which could partially explain why the disease is more severe in obese individuals. (4) HIF effects should be investigated cautiously because it has a little-known range of oxygen sensitivity, which can promote negative and positive effects in the body—depending on how much O_2_ is present and for how long.

Given the hypotheses raised, it is clear how important basic research is to promote answers about a molecule of great significance in oxygen metabolism (HIF) in the context of COVID-19, a primarily pulmonary disease that rapidly devastated the planet. For this, it is suggested that fundamental research studies use the arsenal of experimental methods in congruence with the speed at which the HIF is altered and with the certainty that hypoxia will be present, even in small amounts, in the analyzed samples. This would reduce sampling bias, which may be one of the reasons why studies present such divergent results, and would allow greater robustness in the interpretation of the data. Based on this, studies may have data that support future investigations of therapeutic targets, which aim at direct and/or indirect action on HIF.

## 5. Conclusions

This study’s critical analysis highlights the key role of the hypoxia-inducible transcription factor HIF-1α in the pathophysiology of COVID-19, emphasizing its dual influence on inflammatory processes, metabolic adaptations, and post-acute complications. The scoping review reveals that the activation of HIF-1 α during SARS-CoV-2 infection is intrinsically linked to mechanisms such as silent hypoxia, immune dysregulation, and vascular remodeling, which contribute to the disease’s clinical heterogeneity.

The compiled data demonstrate that, although HIF-1α promotes initial adaptive responses to hypoxia—such as angiogenesis via VEGF and anaerobic glycolysis—its prolonged or dysregulated activation exacerbates the cytokine storm, endothelial dysfunction, and viral persistence. This duality is particularly evident in the modulation of virus entry: while the suppression of *ACE2* and *TMPRSS2* by HIF-1α reduces infectivity in lung epithelial cells, its activation in cardiovascular tissues can increase susceptibility to injury. Furthermore, HIF-1α-mediated metabolic reprogramming in monocytes—characterized by glycolytic dominance and mitochondrial dysfunction—creates a microenvironment conducive to viral replication, leading to worse clinical outcomes [7,9,10,37].

Interventional studies involving HIF-PHD inhibitors, such as Vadadustat, indicate modest benefits in reducing the reliance on ventilatory support; however, these findings do not reach the predefined threshold of statistical significance [88]. Concurrently, antiangiogenic therapies (for example, bevacizumab) present themselves as promising strategies, yet they remain to be validated in extensive cohorts [91]. The intricate nature of the HIF pathway, characterized by its tissue-specific regulation and interactions with other pathways such as NF-κB and NLRP3, emphasizes the necessity for precise therapeutic approaches that consider the timing of intervention and the individualized molecular profile [48,79].

In summary, modulating HIF serves as a viable therapeutic target for mitigating both acute and chronic complications of COVID-19. However, its application necessitates balancing the suppression of detrimental effects with the preservation of adaptive responses. Advances in this field rely on integrating multilevel data—from in silico modeling to functional assays—to translate molecular complexity into clinically effective and safe interventions.

## Figures and Tables

**Figure 1 ijms-26-04202-f001:**
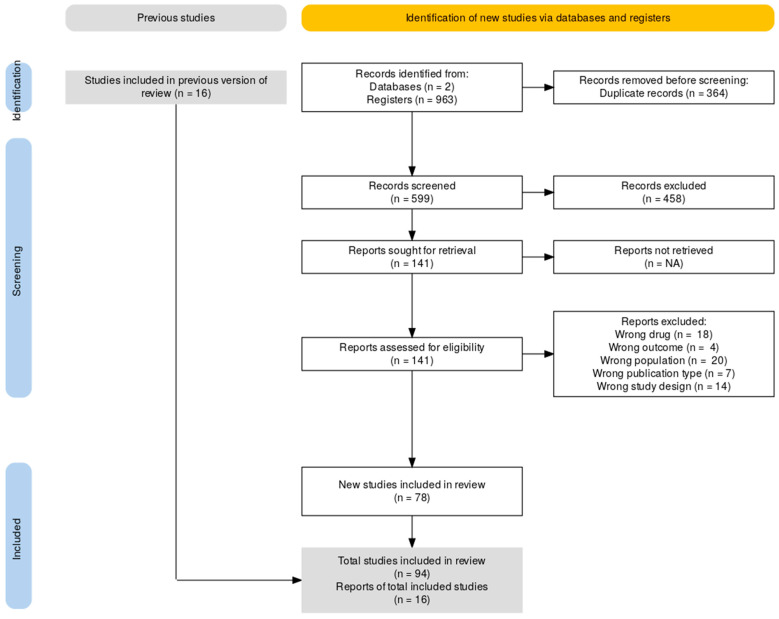
Flowchart following PRISMA-ScR.

**Figure 2 ijms-26-04202-f002:**
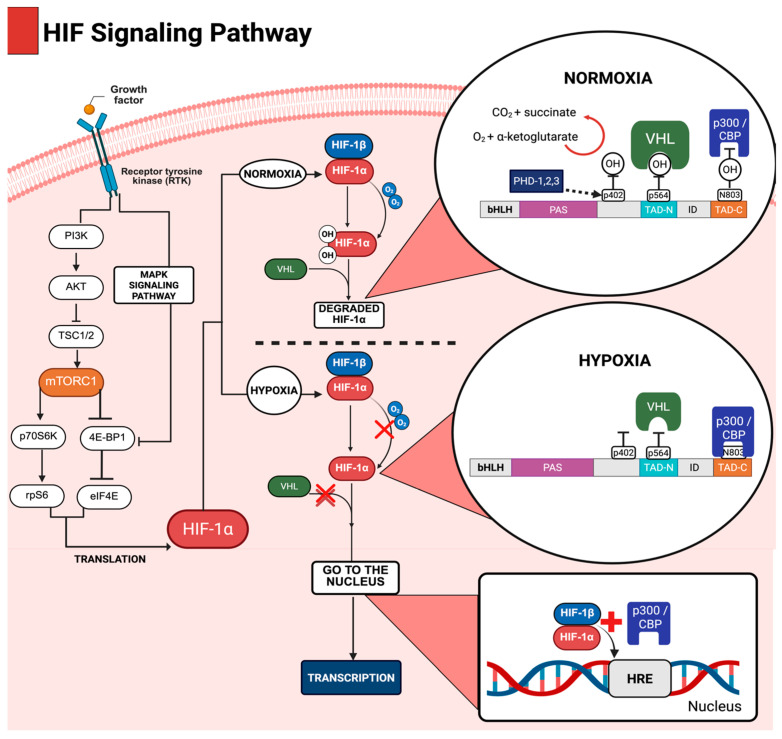
The HIF pathway. Under normoxia, HIF-1α, the oxygen-sensitive subunit of the HIF heterodimer, experiences rapid proteasomal degradation. This process is facilitated by prolyl hydroxylase domain-containing enzymes (PHD1–3), which hydroxylate specific proline residues (Pro402/564 in HIF-1α) in an O_2_-dependent manner. Hydroxylation enables recognition by the Von Hippel–Lindau (VHL), a component of the E3 ubiquitin ligase complex that marks HIF-1α for ubiquitination and subsequent degradation. In contrast, PHD activity is diminished under hypoxia due to limited O_2_ availability, leading to the stabilization of HIF-1α. The stabilized subunit dimerizes with its constitutively expressed partner, HIF-1β and recruits transcriptional coactivators (p300/CBP) to hypoxia response elements (HREs), becoming activated to make the gene transcription.

**Figure 3 ijms-26-04202-f003:**
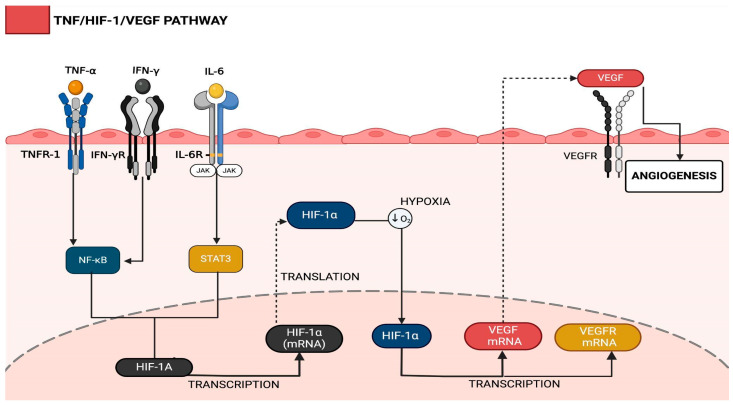
The intersection of the angiogenesis process with the primary immune response. (1) The SARS-CoV-2 virus, upon entering the lung environment, binds to angiotensinogen-converting enzyme 2 (ACE-2) receptors, promoting an increase in angiotensin-2 concentration at the site due to competition for the ACE-2 receptor binding site. In response to the entry of the virus, macrophages of the M1 phenotype secrete pro-inflammatory cytokines, such as TNF-alpha, which bind to TNFR-1 receptors. Consequently, an intracellular signaling response occurs, which releases NF-kB from the inhibitory molecule IKB, allowing NF-kB to become active and enter the nucleus for its transcription action. (2) NF-kB acts by binding to the initiation site of the DNA molecule precursor of *HIF-1α*, inducing the transcription of HIF-1α RNA, which will then be translated into its protein form. In contact with the hypoxic environment, HIF-1α is not degraded and can enter the nucleus to act on the transcription of vascular endothelial growth factor RNA (VEGF-RNA) and its receptor, VEGFR-1-RNA. (3) VEGFR-1-RNA and VEGF-RNA, after being translated into the cytoplasm, will be able to exert the functions of membrane receptors and signaling proteins. VEGF will be secreted from the cell to act on similar cells or even on the cell that produced it.

**Figure 4 ijms-26-04202-f004:**
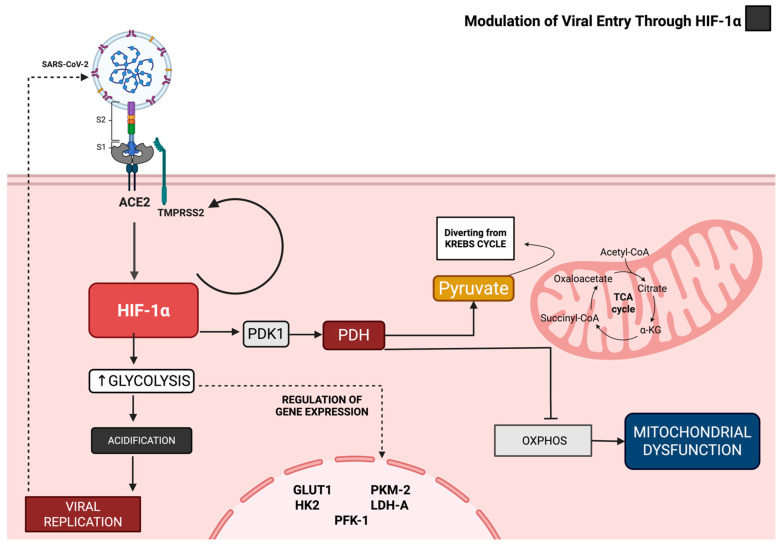
Modulation of Viral Entry Through HIF-1α. ACE2—Angiotensin-Converting Enzyme 2; TMPRSS2—Transmembrane Serine Protease 2; HIF-1α—Hypoxia-Inducible Factor 1-alpha; PDK1—Pyruvate Dehydrogenase Kinase 1; PDH—Pyruvate Dehydrogenase; GLUT1—Glucose Transporter 1; *HK2*—Hexokinase 2; PFK-1—Phosphofructokinase-1; PKM2—Pyruvate Kinase M2; LDH-A—Lactate Dehydrogenase A; TCA—Tricarboxylic Acid (Krebs Cycle)); OXPHOS—Oxidative Phosphorylation.

**Table 1 ijms-26-04202-t001:** This table lists articles employing experimental methods (in vivo, in vitro, in silico), excluding reviews. Double-cited articles utilize multiple experimental methods. HIF (Hypoxia-Inducible Factor): a family of heterodimeric proteins that act as transcriptional factors involved in the cellular response to hypoxia. HIF-1α: HIF isoform activated in a great number of cell types under hypoxic conditions. HIF-2α: HIF isoform associated with a more restricted expression pattern. The nomenclatures of the isoforms were cited in this article as they were originally mentioned in the publications used as references for this paper. Non-human samples utilized for experimental testing were specified.

Methods	Relationship with HIF	Reference
Immunohistochemistry	HIF-1α upregulation confers cytoprotective responses in endothelial cells in the hearts of COVID-19 patients. Human heart tissue samples	Wang et al., 2022 [15]
	*HIF-1α* was equally expressed in COVID-19 patients and the control group. The overexpression of HIF-1α was associated with greater transcription of *VEGF*. Human lung tissue samples	Miggiolaro et al., 2023 [14]
	*Carbonic anhydrase IX (CAIX)* expression represents an indirect indicator of the upregulation of HIF-1 in human villous trophoblastic and stromal cells. Human placenta samples	Mao et al., 2022 [25]
	COVID-19 patients exhibited spatial heterogeneity of Angiopoietin-2, HIF-1α, and TGF-β as indicators of microischemia in affected lung tissue samples compared to healthy controls. Human lung tissue samples	Ackermann et al., 2022 [26]
	HIF-1α transcriptionally upregulates *ACE2* expression. Mouse and human bronchioles samples	Liu et al., 2021 [5]
ELISA	The paper’s analysis did not reveal any significant difference in the plasma level of transcription factor HIF-1α. Human plasma samples	Krenytska et al., 2023 [27]
	HIF-1α upregulation in hypoxia promotes the transcription of *Cygb*, which is produced to supply oxygen in tissues during hypoxic conditions like SARS-CoV-2 infection. Human nose–throat swab samples	Wulandari et al., 2023 [28]
PCR	ACE-2 is regulated by miR-421-5p, leading to the development of immunothrombosis; ⁠miR-421-5p also acts on hypoxia response repressor elements (HRR), resulting in an inflammatory imbalance mediated by the overexpression of HIF and its genes, which results in increased intensity and lung damage. Human serum samples	Abdolari et al., 2022 [29]
	Virus-induced HIF-1α activation leads to increased expression of Fetuin-A, which has anti-inflammatory properties and can modulate the immune response. Human serum samples	Alghanem et al., 2023 [30]
Multi-omics	SARS-CoV-2 modulated the AKT/mTOR/HIF-1α pathway, which regulates glycolysis and glutamatelysis, consequently inhibiting SARS-CoV-2 replication. Mouse lung tissue samples	Ardanuy et al., 2023 [4]
	HIF-1α pathway is upregulated in particulate matter exposed endothelial cells. Human endothelial cell samples	Manivannan et al., 2021 [31]
	Targeting HIF-1α could help mitigate hypoxia-related damage or inflammatory effects in conditions like PACS. Human serum samples	Wang et al., 2023 [32]
In Silico	PKA-inducible HIF-1α was shown to increase coagulation factors and thrombus formation. Therefore, targeting PKA modulation, promoting HIF-1α downregulating, should be considered as a COVID-19 therapeutic. Protein–Protein Interactions (PPIs): human samples	Barman et al., 2022 [33]
	Cytokine storm in SARS-CoV-2 infected lung tissue may be due to the HIF-1α-regulated overexpression of *ACE2* and *TMPRSS2*. Human samples: lung cancer cell line, breast cancer cell line, colorectal adenocarcinoma cell line	Boopathi et al., 2023 [2]
	The HIF-1α signaling pathway is significantly affected by SARS-CoV-2 infection, as indicated by the enrichment of human hub proteins in this pathway. PPIs: human samples	Ghosh et al., 2021 [34]
	The glycolysis/gluconeogenesis and HIF-1α signaling pathways are shown to be associated with COVID-19 and neurological diseases. PPIs: human samples	Rahman et al., 2021 [35]
	HIF-1α plays a role as a transcriptional regulator of the adaptive response to hypoxia, tumorigenesis and metastasis, based on human genes targeted by SARS-CoV-2 encoded miRNAs. Human lung epithelium samples	Roy et al., 2021 [36]
	In COVID-19, the angiogenesis process, stimulated by HIF-1α, is accelerated by MMP (Matrix Metalloproteinases) and NRP (Neuropilins) cooperation, leading to significant tissue damage. Human samples	Saleki et al., 2024 [37]
	*SLC2A3* encodes the glucose uptake transporter GLUT3 and LCP1 (L-Plastin). They are induced during hypoxia by STAT-3-HIF-1α signaling and regulate macrophage infiltration. Human samples	Sheerin et al., 2022 [38]
	HIF-1α upregulation activates both SARS-CoV-2 infection and inflammatory response and plays a role in aggravation of COVID-19. Human blood cells: g T (CD4 + helper T and CD8 + cytotoxic T), B, and natural killer (NK)	Shen et al., 2022 [7]
	HIF-1α pathway enriches a set of proteins that are linked to both the predisposing diseases and to the endocrine-disrupting chemical. Human samples	Wu et al., 2020 [39]
	In the retina, upregulation of the HIF-1α pathway in SARS-CoV-2 infection promotes the expression of *VEGF*, which stimulates angiogenesis. Human eye tissue samples	Yuan et al., 2021 [40]
	Upregulation of HIF-1α in SARS-CoV-2 infection may lead to cytokine storm. Human cardiomyocyte samples	Zhang et al., 2023 [41]
Transcriptomic Analysis	Hypoxic activation of HIF-1α is related to MAPK, NF-kB and IL-6 signaling, which shows its role in cytokine production. Mice monocyte cell samples	Caldwell et al., 2024 [42]
	SARS-CoV-2 infection leads to hypoxic lung tissue conditions, which triggers the HIF-1α pathway. Rhesus macaques solid organs samples	Du et al., 2023 [43]
	There is a molecular link between HIF-1α and neutrophil degranulation in blood. The correlation was more consistent in altitude-related hypoxia than that in COVID-19 or other respiratory infections. Human neutrophils samples	Lei et al., 2024 [44]
	HIF-1α pathway modulates genes related to early inflammatory response, immune response, and cell signal transduction. It acts as a parental gene of circRNAs and plays biological functions in SARS-CoV-2 infection. Human bronchial epithelial cells	Yang et al., 2021 [24]
Proteomic Analysis	Proteomic pathway analysis in SARS-CoV-2 infected human host cells revealed an upregulation of HIF-1α. Human lung and airway cell samples	Maria et al., 2023 [45]
	Proteomics-based studies have observed that SARS-CoV-2 causes global proteomic changes after 48 h SARS-CoV-2 post-infection, specifically in pathways related to HIF-1α. Human lung tissue samples	Sacoon et al., 2021 [46]
	Upregulation of the HIF signaling pathway and ROS production were gradually enhanced during the disease progression in SARS-CoV-2-infected patients. Human serum samples	Wang et al., 2021 [47]
	HIF-1α pathway modulates genes related to early inflammatory response, immune response, and cell signal transduction. It acts as a parental gene of circRNAs and plays biological functions in SARS-CoV-2 infection. Human lung epithelial cell samples	Wang et al., 2021 [24]
Western blotting	HIF-1α transcriptionally upregulates *ACE2* expression. Mouse and human bronchioles samples	Liu et al., 2021 [5]
	HIF-1α enhances the production of pro-inflammatory cytokines, especially IL-6 and TNF-α. Human bronchial epithelial cell samples	Pooladanda et al., 2021 [48]
	In normoxia, PHD2 degrades HIF-1α and HIF-2α; in hypoxia induced by SARS-CoV-2 infection, HIF level is increased in activated platelets, promoting platelet activation, aggregation, and inflammatory signaling. Human platelet and monocyte cell samples	Shrimali et al., 2021 [49]
Multiplex assay	It was postulated that the return to normoxia after a mild hyperoxia stimulus is sensed as a hypoxic trigger, which induces HIF-1α activation and then VEGF synthesis. Human monocyte cell samples	Keller et al., 2023 [50]
Cytometry Immunophenotyping	B cells seem particularly sensitive to perturbations in oxygenation and HIF activity. Mice B cell samples	Kotagiri et al., 2022 [51]
Flow Cytometry	Ethanol consumption resulted in transcriptional shifts in the immune landscape of the lung. Infiltrating monocytes associated with migration were decreased while inflammatory HIF-1α signaling increased. Mice lung tissue samples	Ardanuy et al., 2023 [4]
TUNEL Assay	Upregulation of HIF-1α transcription factor at 48 h SARS-CoV-2 post-infection compared to 24 h. Mice lung tissue samples	Ardanuy et al., 2023 [4]
Mass Spectrometry	Hydroxyglutaric Acid increases concentrations during times of tissue hypoxia via a HIF-dependent pathway; Hydroxyglutaric Acid upregulation may influence the adaptive immune response to SARS-CoV-2, with reports of accumulation, activation and differentiation of CD8+ T-cells. Human serum samples	Whiley et al., 2024 [52]

**Table 2 ijms-26-04202-t002:** Key genes regulated by Hypoxia-Inducible Factor 1α (HIF-1α). Lists the standardized gene names, their corresponding gene identifiers, and primary biological functions. These genes play vital roles in cellular adaptation to hypoxia, regulating processes such as angiogenesis, metabolism, erythropoiesis, vasodilation, coagulation, and cell survival. Gene nomenclature adheres to the KEGG Pathway Database (hsa04066).

Gene ID	Gene Name	Function
*VEGF*	Vascular Endothelial Growth Factor	Promotes angiogenesis by stimulating new blood vessel formation to enhance oxygen delivery to tissues.
*EPO*	Erythropoietin	Stimulates erythropoiesis, increasing oxygen-carrying capacity in the blood.
*SLC2A1*	Solute Carrier Family 2 Member 1 (GLUT1)	Facilitates glucose transport across the plasma membrane, critical for cellular metabolism, particularly under hypoxia.
*LDHA*	Lactate Dehydrogenase A	Catalyzes the conversion of pyruvate to lactate, allowing continuous ATP production via anaerobic glycolysis.
*PGK1*	Phosphoglycerate Kinase 1	A key enzyme in glycolysis that catalyzes ATP generation through substrate-level phosphorylation.
*NOS2*	Nitric Oxide Synthase 2 (Inducible)	Produces nitric oxide, a signaling molecule involved in vasodilation and cellular responses to hypoxia.
*NOS3*	Nitric Oxide Synthase 3 (Endothelial)	Regulates nitric oxide production in endothelial cells, promoting vasodilation and blood flow.
*HMOX1*	Heme Oxygenase 1	Degrades heme into biliverdin, iron, and carbon monoxide, providing cytoprotective effects against oxidative stress.
*HK1/HK2*	Hexokinase 1/2	Catalyzes the phosphorylation of glucose to glucose-6-phosphate, the first step of glycolysis.
*ALDOA*	Aldolase, Fructose-Bisphosphate A	A glycolytic enzyme that cleaves fructose-1,6-bisphosphate into glyceraldehyde-3-phosphate and dihydroxyacetone phosphate.
*ENO1*	Enolase 1	Catalyzes the conversion of 2-phosphoglycerate to phosphoenolpyruvate in glycolysis.
*PFKFB3/PFKL*	6-phosphofructo-2-kinase/fructose-2,6-biphosphatase3/Phosphofructokinase, Liver Type	Regulates glycolysis through control of fructose-2,6-bisphosphate levels, an allosteric activator of phosphofructokinase-1.
*PDK1*	Pyruvate Dehydrogenase Kinase 1	Inhibits pyruvate dehydrogenase, reducing oxidative metabolism and favoring anaerobic glycolysis.
*TIME1*	TIMP Metallopeptidase Inhibitor 1	Inhibits matrix metalloproteinases, regulating tissue remodeling and angiogenesis.
*ITGB2*	Integrin Subunit Beta 2 (CD18)	Component of β2 integrins, involved in cell adhesion and immune responses.
*CD142*	Coagulation Factor III (Tissue Factor)	Regulates blood coagulation and inflammation in hypoxic conditions.
*TFRC*	Transferrin Receptor	Mediates iron uptake, which is essential for cellular respiration and hemoglobin synthesis.
*FLT1*	Fms Related Receptor Tyrosine Kinase 1 (VEGFR-1)	VEGF receptor is involved in angiogenesis regulation and vascular permeability.
*EGF*	Epidermal Growth Factor	Stimulates cell proliferation and tissue regeneration.
*SERPINE1*	Serpin Family E Member 1 (PAI-1)	Regulates fibrinolysis and contributes to thrombosis under hypoxia.
*ANGPT1*	Angiopoietin 1	Modulates vascular stability and remodeling.
*SINGLE*	TEK Receptor Tyrosine Kinase (Tie-2)	A receptor for angiopoietins, crucial for vascular integrity maintenance.
*EDN1*	Endothelin 1	Potent vasoconstrictor regulated by HIF-1α, involved in blood pressure control.
*NPPA*	Natriuretic Peptide A (ANP)	Regulates fluid-electrolyte balance and blood pressure.
*BCL2*	BCL2 Apoptosis Regulator	Anti-apoptotic protein that promotes cell survival under hypoxic stress.
*CDKN1A/CDKN1B*	Cyclin Dependent Kinase Inhibitor 1A/1B (p21/p27)	Inhibitors of cyclin/CDK complexes, involved in cell cycle control and stress response.

**Table 3 ijms-26-04202-t003:** Studies on interventions using drugs that directly or indirectly affect the HIF pathway.

Study	Drug	Population	Methods	HIF Related Results
Lewis, S. A. et al. (2023) [90]	Alcohol (ethanol)	Vero E6 Cells (to obtain SARS-CoV-2 virus) Samples of Bronchoalveolar Lavage (BAL): Monkeys (n = 11) and Humans (n = 6).	Flow Cytometry Luminex scRNA-Seq. Gene Set Enrichment analysis	Higher HIF-1α levels were found in the BAL of rhesus monkeys and humans after six months of chronic alcohol consumption. The DEGs in myeloid cells, such as alveolar macrophages and monocytes, indicate HIF-1α pathway activation.
VSTAT Trial (2022) [88]	Vadadustat (AKB-6548)	449 adult subjects in five hospitals who were randomized 1:1 to vadadustat 900 mg or placebo once daily orally for up to 14 days while hospitalized	Phase 2, randomized, double-blind, placebo-controlled trial.	Vadadusta is an oral hypoxia-inducible factor prolyl hydroxylase inhibitor (HIF-PHI). The VSTAT study assessed vadadustat’s efficacy against a placebo in severe COVID-19 patients using the NIAID Ordinal Scale (NIAID-OS). The results suggest a therapeutic potential but are insufficient for a definitive conclusion under the pre-established parameters. The safety of the drug was comparable to placebo, with no signs of additional toxicity.
Liu et al. (2020) [91]	Bevacizumab (Avastin)	Adult patients with severe COVID-19, characterized by hypoxemia and radiological evidence of pneumonia (n = 26).	Open-label, single-arm clinical trial (single-arm).	Bevacizumab (Avastin) is a monoclonal antibody that inhibits VEGF, which is regulated by HIF-1α. In severe COVID-19 patients, hypoxemia and inflammatory stress stabilize HIF-1α, leading to increased VEGF transcription, which contributes to angiogenesis, vascular permeability, and pulmonary edema. Specifically, the administration of bevacizumab to patients with severe COVID-19 led to a significant improvement in oxygenation parameters and the resolution of pulmonary infiltrates, suggesting that the inhibition of VEGF—a transcriptional target of HIF1α—reduces vascular permeability inflammation.

HIF-1α: hypoxia-inducible factor 1-alpha; VEGF: vascular endothelial growth factor; DEG: differentially expressed genes.

## Data Availability

No new data were created or analyzed in this study. Data sharing is not applicable to this article.

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
