# Peer review of "HIF-1α Pathway in COVID-19: A Scoping Review of Its Modulation and Related Treatments"

_ijms, 2025, doi:10.3390/ijms26094202_

Round 1

Reviewer 1 Report

Comments and Suggestions for Authors

This review aims to elucidate the relationship between COVID-19 and HIF signaling. For this purpose, the Authors made extensive use of AI tools. In my opinion, this tool may be considered legal when the Authors disclose it and clearly specify the terms of use, as they did in this manuscript, but the final judgment on this aspect depends on the ethical rules of this journal, which prohibit the use of AI tools in the review process.

This informative and balanced review concerns an emerging issue that may be key to unraveling the not-yet-understood issue of silent hypoxia. The coverage offered by this manuscript is good, but the Authors neglected a few issues that may deserve more attention:

  1. The time variable: although critical, it is never well appreciated that during persisting hypoxia HIF is not constant over time but is subjected to progressive destabilization. Such a time-dependent decrease would be mirrored by decreased downstream effects following different time courses. Very few articles discuss this aspect, but the Authors are nevertheless invited to comment briefly on this effect, which may be linked to the silent hypoxia phenomenon.
  2. The role of HIF-2a, which may be critical for sustained hypoxia, and its effect on erythropoietic, angiogenetic, and cell-matrix changes. The short statement on line 188 is not sufficient to frame this critical issue.
  3. The similarities and differences between COVID-19 and ARDS.
  4. The hyperoxic treatment to which COVID-19 patients are often exposed, with consequent hyper-inflammatory and pro-oxidant state, is at the same time a separate counterintuitive trigger for HIFs rise, a factor that disturbs the redox balance, and in general a source of uncertainty in data interpretation.

Minor

The title highlights the “hypoxia mechanisms” but the manuscript is mainly centered on HIF1a. Please adjust the title accordingly.

Abstract, line 22. Please be more precise with the term “dysregulation”, is it increase or decrease? This term has been used also elsewhere; when possible, specify if it implies an increase or a decrease.

Give more info on the Rayyan software, which may not be known by part of the potential audience.

Figure 1 needs info on how it was obtained. Specifically, the meaning of some connections is questionable, e.g., glucose control, inflammation, matrix function and apoptosis might affect aerobic metabolism as well. Also, clarify what is meant by "independent metabolism". Some of the yellow and red areas do not have an evident explanation.

Line 150, specify which are the “predetermined inclusion criteria”. The reasons for excluding so many reports in Figure 2 need to be highlighted better.

Figure 3, correct “degrated” HIF into “degraded”. Substitute the circles with 1 and 2 with normoxia and hypoxia.

Line 212, is the reference to Figure 1 correct?

The paragraph starting on line 315 is unclear, please rewrite it.

Table 1. Please report also the name of the first Author and year, as in Table 3. Reformat the table to improve readability, e.g., try landscape instead of portrait, further summarize the text without making it unclear, 1st column as vertical text, etc.

Consider removing Table 2, which reports established data not always linked to this manuscript.

The link between the main argument under study and H1N1 is not clear and looks redundant. Remove it or explain why it is essential.

Line 298: I doubt whether data obtained in post-mortem analyses of COVID-19 lungs, that apparently reveal HIF-1α overexpression, are reliable because HIF is a labile protein that undergoes time-dependent changes upon environmental perturbations that lead to death.

There are two “Figure 3”.

Reviewer 2 Report

Comments and Suggestions for Authors

This review article is discussing the interactions between HIF-1α and the pathophysiology of COVID-19, and easy to read. There are several such review article. It is important to describe the originality and the novelty of this article in the introduction. In addition, to improve the content furthermore, please improve following points.

(1) The aim of this article was to discuss the role of HIF-1αin emphasizing its relationship with Post-Acute COVID-19 Syndrome (PACS). However, this part is weak, whereas there are many descriptions about basic knowledge about HIF-1α.

(2) Materials and Methods is poor. The authors described in the section that "the supplementary materials detail the complete list of descriptors used" in line 104. However, there is no list of supplementary materials.

(3) Line 122: Please list all AI tools to use preparation of this review articles.

(4) Fig.2: There is a typo in the figure. What is independent mechanism? Is it oxygen independent metabolism?

(5) How do the authors define record? Only original articles including Note and Letter formats?

(6) Table 2 and others: If the definition of HIF-1α and HIF is different, it should be explained.

(7) Table 2 and others. Please indicate clearly that the data was from humans or mice using which tissue or cellular samples.

(8) Line 174; HIF-α is correct?

(9) Fig. 3: It is better to include Word HREs and show binding of HIF-1αand HIF-1β in figure of nucleus.

(10) Line 187, 368, 370: Please check if the full name of HIF-1α needs to be indicated here.

(11) Line 183-185 and Fig.3: Please indicate which part in Fig. 3 is oxygen-independent pathway.

(12) Line 204-209: Is this regulation of HIF-1α different from that in oxygen-independent pathway?

(13) Line 344: Only TNF-α is important pro-inflammatory mediator?

(14) Section 4.5.1 and 4.5.2: Please include figure(s) showing the content of this section.

(15) Line 503-509: Please include name of experimental drugs and references. It is better to make a table showing information about the drugs targeting HIF-1α directly.

(16) Line 552-557: Future perspective is weak. Please include your hypothesis and describe how basic research and clinical application of HIF-1α and its associated mechanism can be and should be developed, more.  

(17) Line 566-583: Please include references properly.

Round 2

Reviewer 1 Report

Comments and Suggestions for Authors

The authors have done an egregious job of improving their manuscript. My remaining concerns regard two of the raised issues.

Response 2. HIF 1a and HIF 2a also differ in the target genes. In Line 368, delete “main”, as you may mean “the two oxygen-responsive subunits”. As each experimental model is expected to behave differently, report in the text which model and hypoxia severity were tested in ref 70. In Line 387, “it also has variations in oxygen levels since the samples were collected due to the sample analyses” is unclear; please reword that sentence.

Response 5. Note that in several tissues, including brain and heart, hyperoxic treatments led to increased, not decreased, tissue levels of HIF, especially HIF 2a. And that in peri-operative environments, hyperoxia is being reconsidered as potentially dangerous.

Response 9. I agree.

Response 15, ok.

Author Response

Dear Reviewer 1,

Thank you very much for having reevaluated our paper with better grades and for the new suggestions, which increasingly enrich our work. Below are the responses to your comments (Obs.: In the Word file, we highlighted the new changes in blue).

Comments 1: “Response 2. HIF 1a and HIF 2a also differ in the target genes. In Line 368, delete “main”, as you may mean “the two oxygen-responsive subunits”. As each experimental model is expected to behave differently, report in the text which model and hypoxia severity were tested in ref 70. In Line 387, “it also has variations in oxygen levels since the samples were collected due to the sample analyses” is unclear; please reword that sentence.”

Response 1: Thank you for pointing this out. We agree with this comment. We have, accordingly, modified the text in Lines 367 – 378 to correct the word “main”, and to explain the methods used by ref. 70. We also revised and modified the text in Lines 388 – 393 for clarity.

Comments 2:Response 5. Note that in several tissues, including brain and heart, hyperoxic treatments led to increased, not decreased, tissue levels of HIF, especially HIF 2a. And that in peri-operative environments, hyperoxia is being reconsidered as potentially dangerous.”

Response 2: Thank you for pointing this out. We included a new paragraph in Lines 445 – 457 addressing this conflict. This point is complicated to answer because the scientific literature does not have a consensus on it. We have found some papers discussing an increase in HIF-2a, but we also found papers mentioning a decrease in both HIF-2a and HIF-1a. Let me explain what we have done: we researched your point and read it carefully. Afterwards, we added information about HIF-2a increases to the text. The references in our list were updated to include these two studies

Thank you. 

Reviewer 2 Report

Comments and Suggestions for Authors

The authors replied to the comments of this reviewer well and improved the content of manuscript. However, it would be better to amend following points furthermore.

(1) Line 411-439: It can be shorten. The author discussed about one study (reference 51) using two paragraphs.

(2)Line 431-446: Please add references. There is no reference in this paragraph.

Author Response

Dear Reviewer 2, 

We have corrected the text as per your recommendation. You can see below: 

Comments 1: "Line 411-439: It can be shorten. The author discussed about one study (reference 51) using two paragraphs."

Response 1: Thank you for pointing this out. We agree with this comment. We have, accordingly, revised and modified the text in Lines 416 – 428 to shorten it. 

Comments 2: "Line 431-446: Please add references. There is no reference in this paragraph."

Response 2: Thank you for pointing this out. We agree with this comment. We have, accordingly, modified the text in Lines 431, 434, and 444 to add the references. 

Thank you.